

# Silk derived formulations for accelerated wound healing in diabetic mice

Muniba Tariq, Hafiz Muhammad Tahir, Samima Asad Butt,
Shaukat Ali, Asma Bashir Ahmad, Chand Raza, Muhammad Summer,
Ali Hassan and Junaid Nadeem

Department of Zoology, Government College University Lahore, Lahore, Pakistan

## ABSTRACT

**Background:** The present study aimed to prepare effective silk derived formulations in combination with plant extract (*Aloe vera* gel) to speed up the wound healing process in diabetic mice.

**Methods:** Diabetes was induced in albino mice by using alloxan monohydrate. After successful induction of diabetes in mice, excision wounds were created via biopsy puncture (6 mm). Wound healing effect of silk sericin (5%) and silk fibroin (5%) individually and in combination with 5% *Aloe vera* gel was evaluated by determining the percent wound contraction, healing time and histological analysis.

**Results:** The results indicated that the best biocompatible silk combination was of 5% silk fibroin and 5% *Aloe vera* gel in which wounds were healed in 13 days with wound contraction: 98.33 ± 0.80%. In contrast, the wound of the control group (polyfax) healed in 19 day shaving 98.5 ± 0.67% contraction. Histological analysis revealed that the wounds which were treated with silk formulations exhibited an increased growth of blood vessels, collagen fibers, and much reduced inflammation.

**Conclusion:** It can be concluded that a combination of *Bombyx mori* silk and *Aloe vera* gel is a natural biomaterial that can be utilized in wound dressings and to prepare more innovative silk based formulations for speedy recovery of chronic wounds.

## INTRODUCTION

Cutaneous wound healing is a programmed, multifaceted and sequential biological process that relies on the interaction between a large number of cells and molecular factors to repair the barrier function of the skin (*Paquette & Falanga, 2002*; *Martínez-Mora et al., 2012*). A "wound" is the disruption of normal skin physiology, while "wound healing" is the process by which the body pursues to restore skin stability (*Sugihara et al., 2000*). Ideal healing of a skin wound requires synchronized incorporation of all molecular and biochemical events of cell proliferation, migration, deposition of extracellular matrix and remodeling (*Das & Baker, 2016*). However, this orderly advancement of the healing process is compromised in chronic, non-healing wounds (*Falanga, 2005*). Chronic wounds normally occur in diabetic patients due to their impaired wound healing process (*Spampinato et al., 2020*; *Farman et al., 2020*).

Corresponding author
Shaukat Ali,
dr.shaukatali@gcu.edu.pk

Diabetes is a prevalent health challenge that impacts people worldwide. It is recognized as a group of varied disorders with the common elements of glucose intolerance, hyperglycemia caused by insulin shortage, reduced efficacy of insulin action, or both (*Alberti & Zimmet, 1998*; *Atlas, 2015*). Non-healing chronic wounds are considered as one of the most serious complications of diabetes. Such complications are associated with an increased risk of bacterial infection, blood vessel and nerve injury, and ultimately amputation of limbs and other organs (*Masood et al., 2019*).

The wound healing process in diabetic patients is profoundly slow as compared to healthy individuals, hence prolonged healing duration increases the risk of wound associated infections (*Menke et al., 2007*; *Dehghani et al., 2020*).The current wound healing investigations signify the therapeutic potential of formulations in modulating the wound healing process and reducing suffrage of patients (*Nithya, Brinda & Anand, 2011*). Scientists have tried different chemicals and herbal formulations to speed up wound healing in diabetic patients but there were certain limitations and the results were not much persuasive.

There is a long history of utilization of natural materials in the biomedical industry. Amongst many naturally occurring materials, silk obtained from silkworms is considered as an exceptional biomaterial which has a wide range of medical applications (*Jastrzebska et al., 2015*; *Farokhi et al., 2018*; *Tahir et al., 2020*; *Elahi et al., 2020*). It is classified as a 'model biomaterial' due to its remarkable mechanical strength (*Vollrath & Porter, 2006*; *Tahir et al., 2019*), impressive biocompatibility with skin tissues, negligible immunogenicity (*MacIntosh et al., 2008*) and minimal bacterial adhesion (*Cassinelli et al., 2006*). Mounting evidence of preclinical research demonstrates excellent wound healing properties of silk proteins since the 1990s (*Shailendra & Das, 2019*). Silk (particularly silkworm silk) started its journey in the biomedical industry when it was first used to suture skin wounds (*Altman et al., 2003*).

Silk is the strongest and most flexible naturally occurring fiber. It is smooth, shiny and soft in texture unlike most of the synthetic fibers (*Altman et al., 2003*). There are two proteins fibroin (80%) and sericin (20%) present in the silk thread which is secreted by *Bombyx mori's* silk glands (*Vepari & Kaplan, 2007*; *Kundu et al., 2014*; *Aramwit, Siritientong & Srichana, 2012*; *El-Fakharany et al., 2020*). The middle and posterior silk glands of *B mori* larvae produce a fibroin layer and three layers of sericin respectively (*Zhou et al., 2000*). Sericin and fibroin play an active role in accelerating wound healing (*Li et al., 2020*). The wound healing potential of sericin in cell culture and animal models is well reported (*Aramwit & Sangcakul, 2007*). Successful trials with fibroin based biomaterials, for example, sponges (*Roh et al., 2006*), hydrogels (*Fini et al., 2005*), films (*Sugihara et al., 2000*) and nanofibers mats (*Schneider et al., 2009*) have been conducted with impressive results. It has also been reported that silk based wound dressings stimulate cell proliferation and recruitment of cells such as keratinocytes in the wound bed to accelerate the wound healing process (*Chouhan & Mandal, 2020*). Scientists have prepared, silk fibroin/keratin-based biofilms to control the release rate of elastase enzyme in the chronic wound milieu (*Roh et al., 2006*; *Vasconcelos et al., 2010*).

As silk fibroin and sericin exhibit unique biological and physical properties, they are extensively explored by researchers in the biomedical industry for their utilization in wound healing materials. The current study attempts to evaluate the silk-based formulations in treating induced-skin wounds in diabetic mice model. The objectives of this study were to extract pure silk fibroin and silk sericin from silkworm cocoons and to prepare silk-based formulations in combination with plant extract that is, *Aloe vera* gel. Furthermore, in vivo wound healing potential of silk based formulations in artificially wounded diabetic mice model was also evaluated.

## MATERIALS AND METHODS

### Ethical statement

All animal trial techniques were directed as per local and worldwide controls. The nearby direction is the Wet op de dierproeven (Article 9) of Dutch Law (International) as detailed in our previous studies (*Ali et al., 2020a*; *Ali et al., 2020b*; *Hussain et al., 2020*; *Ara et al., 2020*; *Ali et al., 2019*; *Khan et al., 2019*; *Mumtaz et al., 2019*; *Mughal et al., 2019*; *Dar et al., 2019*) and The Institutional Bioethics Committee at Government College University Lahore, Pakistan (No. GCU/IIB/21 dated: 08-01-2019).

### Rearing of mice in animal house

The Swiss albino mice weighing around 29–30 g and 8 weeks old were obtained from the Animal House, Department of Zoology, Government College University Lahore, Pakistan and used as experimental models. They were reared in standard plastic cages of length 10 inches, height 7 inches and width 5 inches in the same Animal House facility of Zoology Department, Government College University Lahore. Six mice were reared per cage under standard laboratory conditions (temperature 19–21 °C, humidity 45–65% and 12 hr light-dark cycle). They were fed standard animal diet and tap water in the cage. Mice were acclimatized for one week before the experimental procedures. The weight of each mouse was measured and noted throughout the experiment.

### *Diabetes induction*

Alloxan and streptozotocin both are widely used diabetogenic agents, but alloxan was preferred over streptozotocin because it was easily available here at low cost. A single dose of alloxan monohydrate (CAT A7413-10G, Signa–Alrich, Germany) was injected intraperitoneally to induce type 1 diabetes. The dosage of alloxan monohydrate was freshly prepared in saline solution at a dosage of 200 mg/kg body weight (*Ahmadi et al., 2012*). All animals were fed with glucose solution (10%) after receiving an injection of alloxan monohydrate to prevent them from sudden hypoglycemic state (*Vanitha & Karthikeyan, 2013*; *Bouzghaya, Amri & Homblé, 2020*). After 24 h of induction of diabetes, blood samples were collected by pricking the tail tip of the mice. Blood glucose level was measured with an electronic glucometer (On-call extra blood glucose meter and test strips). Animals with a blood glucose level of ≥250 mg/dl were considered diabetic and were selected for further experimentation (*Chen et al., 2015*). Mice were given free access to food and water during the study and they were kept in standard plastic cages at room

temperature in the Animal House facility. Blood glucose levels of all albino mice were recorded before starting of the experiment. Only those mice that have normal blood glucose levels were used for further study and those having high blood glucose levels were excluded from the study (*Dra et al., 2019*).

## Creation of skin excision wound in mice

Mice were randomly divided into six groups with each group consisting of 6male mice. Animals were anesthetized intraperitoneally with ketamine (100 mg/kg) and Xylazine (10 mg/kg) in saline before wound induction. The dorsal fur of mice was shaved completely by using an electrical hand shaver. Two full thickness excision wounds were created on the dorsum of each mouse by using a 6mm biopsy punch device. These surgical interventions were carried out under sterile conditions. The total surgical time was 15–20 min for each mouse. All animals received their respective treatments once a day from post wounding day till complete healing. Body weight, skin color and skin irritation were observed and recorded daily.

## Extraction of sericin from cocoons

Silk cocoons of *B. mori* (silkworm) were kindly supplied by the Sericulture section of Forestry department, Punjab, Pakistan. These cocoons were sliced into small pieces. For sericin extraction, 5 g of silk cocoon pieces were immersed in 100 ml of distilled water and autoclaved at 121 °C and 15 lb per square inch pressure for 1 hr. After 1hr the sericin solution was allowed to cool at room temperature and then filtered through a filter paper. The filtration process removed impurities from the sericin solution. The filtered sericin solution was subjected to lyophilizer (freeze drying) at −82 °C for 72 hr to obtain sericin powder (*Martínez et al., 2017*). Extraction procedures of sericin and fibroin were carried out separately by utilizing fresh silk cocoons each time.

## Extraction of fibroin
### Degumming

Silk cocoons synthesized by *B. mori* silkworms were soaked in warm water to loosen the threads. Silk threads from several cocoons were then unwound to obtain silk fibers. Raw silk fibers were then degummed in 0.5% $NaHCO_3$ at 100 °C for 1 hr rinsed thrice with distilled water and then dried overnight in oven (60–80 °C) (*Ju et al., 2016*; *Tahir et al., 2020*).

### Dissolution of silk fibers

Degummed silk (80 mg) was dissolved for 6–8 hr at 80 °C with constant stirring in a solvent system of calcium chloride: ethanol: distilled water in a molar ratio of 1:2:8 (*Wang & Zhang, 2013*; *Yi et al., 2018*). Urea (8 mM) was also added to calcium chloride solvent to achieve 100% dissolution of silk fibers (*Min et al., 2004*).

### Dialysis

After dissolution, the remnants of chemicals were removed through dialysis with a cellulose dialysis membrane in distilled water for 3 days. After dialysis the silk fibroin

solution was sonicated at 20 kHz: 400W for 1 hr and then lyophilized to obtain silk fibroin powder (*Ha, Park & Hudson, 2003*; *Siavashani et al., 2020*).

## SEM analysis of silk fibroin and silk sericin

Powdered samples of silk fibroin and silk sericin were subjected to SEM (Scanning Electron Microscopy) (FEI NOVA 450 Nano SEM) (voltage 1000 kV) available at LUMS (Lahore University of Management Sciences). SEM analysis was done to estimate the approximate sizes of silk fibroin and sericin particles.

## Extraction of *Aloe vera* gel

Fresh *Aloe vera* gel was extracted from the leaves of the plant. The pulp was scraped out from the leaves and blended into a smooth paste using a high-speed blender. The extracted gel was transferred into an airtight container and refrigerated (4 °C). This extraction was carried out under sterile conditions.

## GC-MS analysis of *Aloe vera* gel

*Aloe vera* gel (5 ml) extracted from the leaves was analyzed by GC-MS (Gas chromatography-Mass spectrometry) on a GC-MS equipment at Department of Chemistry, GC University Lahore. GC-MS analysis was performed to detect the bioactive compounds present in the *Aloe vera* gel. The parameters used in GC-MS analysis were Retention time (RT), I Time, F Time, Area, Area%, Height, Height%, A/H and Base (m/z).

## Preparation of formulations

Gel formulations were prepared for four treatment groups. There were two control groups that is, positive control in which wounds were treated with polyfax (Polyfax is a skin ointment with active ingredients Bacitracin zinc and Polymiyxin B sulphate. Both of these ingredients are antibacterial. This ointment is used for the treatment of infected surgical cuts, burns, infected wounds, infected ulcers on skin etc) and negative control in which wounds were washed with saline solution (0.9%) daily. All the groups are shown below:

**T1** 5% Sericin
**T2** 5% Sericin and 5% *Aloe vera* gel
**T3** 5% Fibroin
**T4** 5% Fibroin and 5% *Aloe vera* gel
**C1** Positive control (Polyfax)
**C2** Negative control (Saline solution)

### *Sericin (5%)*

The gel was prepared by dissolving 0.1g sodium carboxy-methyl-cellulose Na-CMC in distilled water to form a homogenous solution. Sericin solution (5%) was prepared in distilled water. Sericin solution was added to the Na-CMC solution with constant stirring until it became a homogenous gel (*Ersel et al., 2016*; *Nishida et al., 2011*).

### *Fibroin (5%)*

Fibroin gel was also prepared by adopting the method outline above. Na-CMC (0.1 g) was dissolved in distilled water to form a homogeneous solution. The fibroin solution (5%) was

prepared in distilled water and added to the Na-CMC solution with constant stirring until the solution became thick and homogenous (*Nishida et al., 2011*).

### Sericin (5%) and Aloe vera gel (5%)

Sericin solution (5%) was prepared in distilled water, mixed with 5% *Aloe vera* gel and vortexed for 1 min. The solution was stored in falcon tubes at low temperature (4 °C) to prevent the growth of microorganisms.

### Fibroin (5%) and Aloe vera gel (5%)

Fibroin solution (5%) was prepared in distilled water and mixed with 5% *Aloe vera* gel. The solution was vortexed for 1 min and stored at low temperature (4 °C) in falcon tubes.

## Application of gel formulations on wounds

The diabetic mice were subjected to their respective treatments till complete wound healing. The formulations were applied evenly on the wound surface daily.

## Percent wound contraction

After wound creation, the wound margins were traced at 2 days interval on transparent graph paper. Measurements were continued until the complete (98–99%) wound restoration. After 2 days interval, the healed area was calculated. The contraction was represented as percent wound contraction and epithelialization time was observed after complete healing (*Lodhi et al., 2016*).

The rate of healing as percentage contraction was calculated using the formula:

$$= \frac{\text{Initial wound area-Wound area on a specific day}}{\text{Initial wound area}} \times 100$$

## Histological evaluations

Skin sample of one mouse from each group was acquired at post wounding day 5 and 10. The central portion of tissue was fixed in 10% buffered formalin (pH = 7). Thin sections were prepared using a microtome and stained with hematoxylin-eosin and Masson's trichrome method. Wound healing effects were examined histologically under a light microscope using low power magnification (*Aramwit & Sangcakul, 2007*).

## Euthanization and Dissection of animals

For euthanization, mice were placed in beakers and euthanized with a large piece of cotton soaked in chloroform. Beaker was covered properly with an aluminum foil. The mice were euthanized within 10–15 min. All euthanized mice were dissected and then skin samples ere collected for histological evaluation.

## Statistical evaluations

For statistical analysis, the normality of the data was assessed using Shapiro–Wilk's test. One-way ANOVA was conducted out to compare percent wound contractionin control and treatment groups, followed by Tukey's post-hoc test using SPSS software (version 20). All data were expressed as the mean ± SEM (Standard Error of Mean).

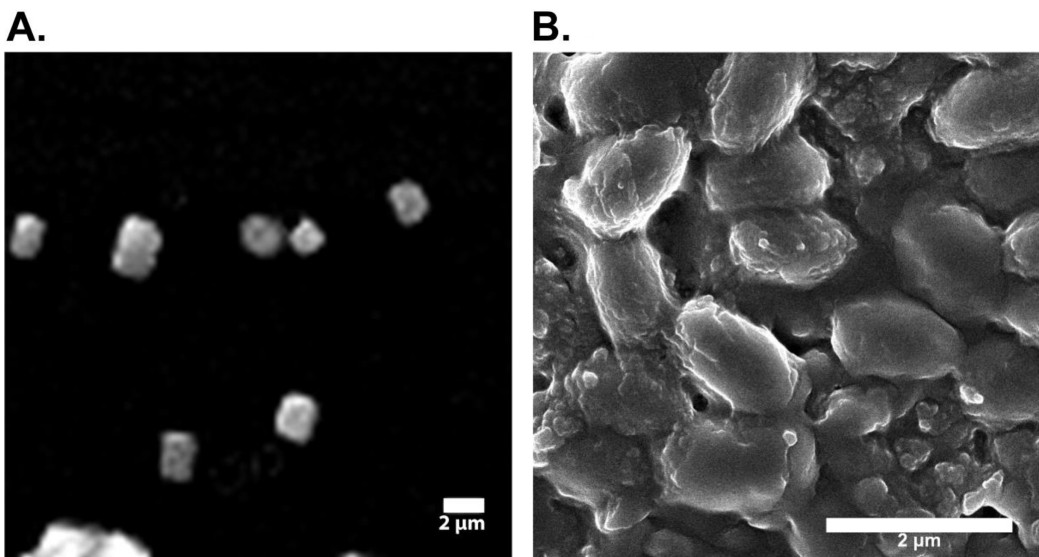

**Figure 1 Electron micrographs of silk fibroin and sericin.** (A) Electron micrograph of sonicated silk fibroin. (B) Electron micrographic image of silk sericin.

## RESULTS

### SEM analysis of silk fibroin and silk sericin

The scanning electron micrographs (SEM) showed 240–300 nm sized silk fibers of sonicated samples (Fig. 1). SEM micrographs of silk sericin at 2 μm scale bar are shown in Fig. 1. Results of SEM indicated that the size of the of silk sericin particles is approximately 102.5 nm.

### GC-MS analysis of *Aloe vera* gel

A total of seventeen compounds were detected in *Aloe vera* gel by GC-MS analysis (Table 1). Five major compounds (2,4:3,5:6,7-Tri-O-benzylidene-1-deoxy-d-gluco-d-gulo-heptitol, stannane bis diphenyl, isopropyl myristate, 9-Octadecenoic acid and 10-Octadecenoic acid)identified in *Aloe vera* gel. Their molecular formula, molecular weight (MW), retention time (RT) and peak area (%) are presented in Table 2. Detail of major and minor compounds (Tables 1 and 2) detected through GC-MS analysis of *Aloe vera* gel will be helpful in future wound healing studies and they may be utilized individually or in combinations for preparing more effective gel formulations to treat chronic wounds.

### Assessment of wound contraction

Healing area of wounds in treatment and control groups at day 11 is presented in Fig. 2 and at different days as percent wound contraction in Fig. 3.

#### Percent wound contraction at various days

Overall, there was significant difference in percent wound contraction between the treatment and control groups at day 3 ($F_{5,30} = 3.391$; $P = 0.015$), day 7 ($F_{5,30} = 7.561$; $P < 0.001$) and day 11 ($F_{5,30} = 29.19$; $P < 0.01$). There is a non-significant variation in

**Table 1 List of major and minor compounds detected through the GC-MS analysis of *Aloe vera* gel.**

|  | Compound name | Molecular formula | Molecular weight |
|---|---|---|---|
| 1 | 2,4:3,5:6,7-Tri-O-benzylidene-1-deoxy-d-gluco-d-gulo-heptitol | $C_{28}H_{28}O_6$ | 460 |
| 2 | Glycine | $C_{36}H_{69}NO_6Si_3$ | 695 |
| 3 | Di-1,3-xylyl-24-crown-6, 5,5'-dimethyl-2,2'-bis(2-propenyloxy) | $C_{32}H_{44}O_8$ | 556 |
| 4 | Decyl .alpha.-d-galactoside, 2,4,6-detrioxy-3-O-benzyl-4,6-S-dibenzylthio | $C_{37}H_{50}O_3S_2$ | 606 |
| 5 | 1,5-Anhydro-2,3-dibenzoyl-4,6-O-dibenzyl-d-glutitol | $C_{34}H_{32}O_7$ | 552 |
| 6 | Colchicine | $C_{31}H_{31}NO_7$ | 529 |
| 7 | Stannane, bis (pentafluorophenyl) diphenyl | $C_{24}H_{10}F_{10}Sn$ | 608 |
| 8 | Inositol | $C_{24}H_{60}O_6Si_6$ | 612 |
| 9 | Galactonic acid | $C_{24}H_{60}O_7Si_6$ | 628 |
| 10 | Myo-Inositol | $C_{24}H_{60}O_6Si_6$ | 612 |
| 11 | Isopropyl Myristate | $C_{17}H_{34}O_2$ | 270 |
| 12 | 9-Octadecenoic acid | $C_{21}H_{38}O_4$ | 354 |
| 13 | Dodecanoic acid | $C_{15}H_{30}O_2$ | 242 |
| 14 | Hexadecanoic acid, methyl ester | $C_{17}H_{34}O_2$ | 270 |
| 15 | 10-Octadecenoic acid | $C_{19}H_{36}O_2$ | 296 |
| 16 | Pentadecanoic acid, 14-methyl-, methyl ester | $C_{17}H_{34}O_2$ | 270 |
| 17 | 12-Octadecenoic acid, methyl ester | $C_{19}H_{36}O_2$ | 296 |

**Table 2 List of five major compounds with their retention time (RT) and peak area (%) detected through the GC-MS study of *Aloe vera* gel.**

| No | RT | Name of the compound | Molecular formula | Molecular weight | Peak area (%) |
|---|---|---|---|---|---|
| 1 | 13.47 | 2,4:3,5:6,7-Tri-O-benzylidene-1-deoxy-d-gluco-d-gulo-heptitol | $C_{28}H_{28}O_6$ | 460 | 10.83 |
| 2 | 13.443 | Stannane, bis (pentafluorophenyl) diphenyl. | $C_{24}H_{10}F_{10}Sn$ | 608 | 10.77 |
| 3 | 17.879 | Isopropyl Myristate | $C_{17}H_{34}O_2$ | 270 | 15.98 |
| 4 | 18.894 | 9-Octadecenoic acid | $C_{21}H_{38}O_4$ | 354 | 23.90 |
| 5 | 20.531 | 10-Octadecenoic acid | $C_{19}H_{36}O_2$ | 296 | 38.52 |

percent wound contraction between T1 (5% sericin) and C1 (positive control; polyfax) ($P > 0.05$ ANOVA) at day 3. However, there was a significant difference in percent wound contraction on day 3 between T3 (5% fibroin) and C1 (positive control; polyfax) ($P = 0.043$ ANOVA).

At day 7 results of Tukey's test indicated that group C1 (positive control; polyfax) differs non-significantly from T1 (5% sericin) ($P > 0.05$ ANOVA) and T2 (5% sericin and 5% *Aloe vera*)($P > 0.05$ ANOVA).On the other hand, there was a significant difference between T3 (5% fibroin) and C1 (positive control; polyfax) ($P = 0.037$). At day 11 results of Tukey's test showed that group C1 (positive control; polyfax) differs significantly from T3 (5% fibroin) ($P = 0.013$ ANOVA) and T4 (5% fibroin and 5% *Aloe vera* gel) ($P < 0.01$ ANOVA) in percent wound contraction (Fig. 3). However, the group C1 (positive control; polyfax) differs non-significantly from T1 (5% sericin) ($P > 0.05$ ANOVA) and T2 (5% sericin and 5% *Aloe vera*) ($P > 0.05$ ANOVA).

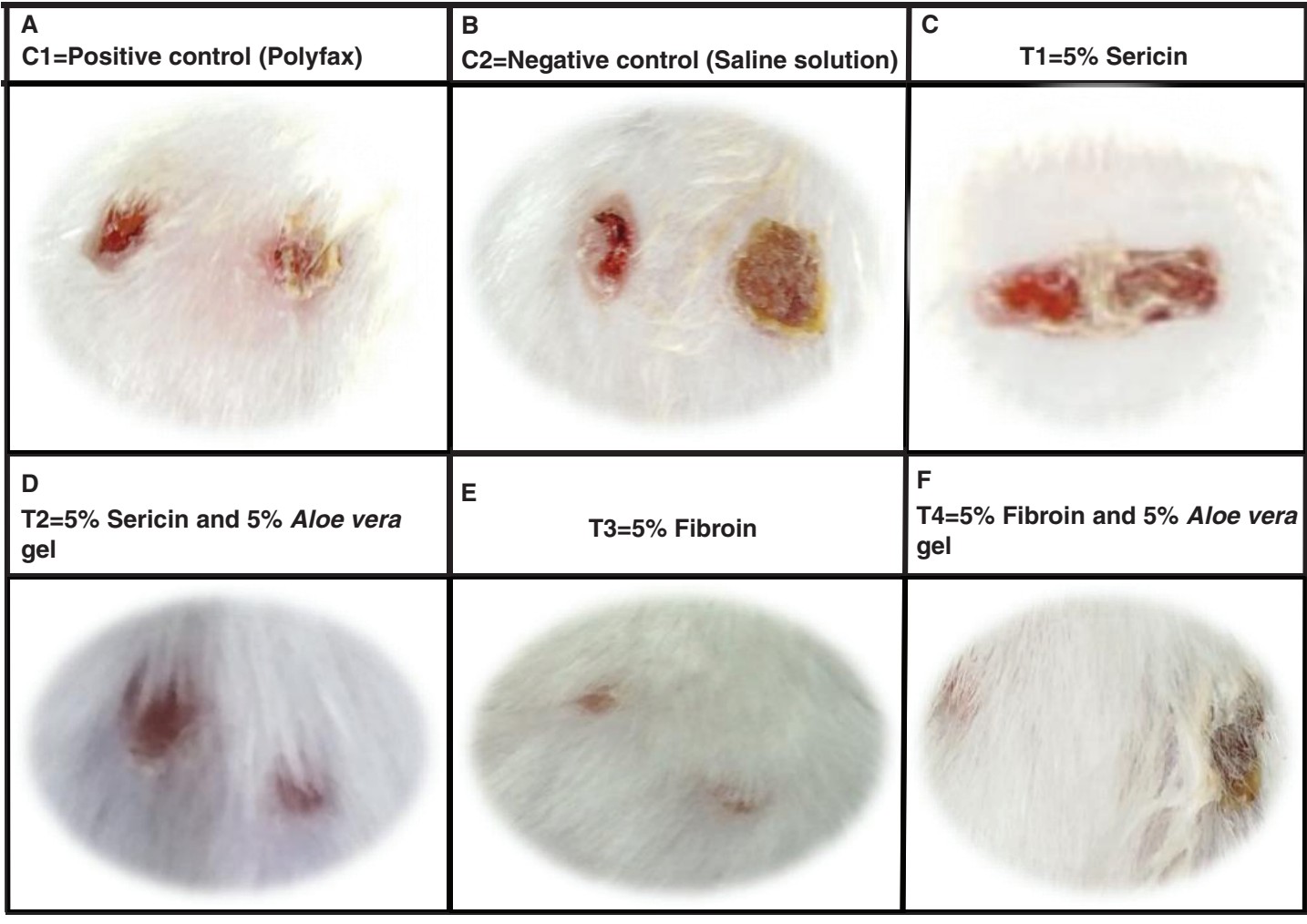

**Figure 2 Wound healing process in different treatment groups at post wounding day 11.** (A) C1 = Positive control (Polyfax); (B) C2 = Negative control (Saline solution); (C) T1 = 5% Sericin; (D) T2 = 5% Sericin and 5% *Aloe vera* gel; (E) T3 = 5% Fibroin; (F) T4 = 5% Fibroin and 5% *Aloe vera* gel.

## Histological analysis

Images of wound size in different treatment and control groups at post wounding day 10 is shown in Fig. 4. Best histological results were observed in group T4 (5% fibroin and 5% *Aloe vera* gel) in which the formation of the new epidermis was initiated and dermis with blood vessels and hair follicles were observed at post wounding day 10. However, histological results from group C1 (positive control; polyfax) showed the formation of collagen fibers and formation of thin epithelium and dermis at post wounding day 10. Healing of wound was incomplete until day 10 in positive control. In the group C2 (negative control; saline solution) there were inflammatory cells and adipose tissues at post wounding day 10.

Histological examination of wounded tissues from group T1 (5% sericin) showed the formation of the new epithelial layer. The wound was not completely epithelialized till day 10 and inflammatory cells were also observed. The histology of wound at 100X of

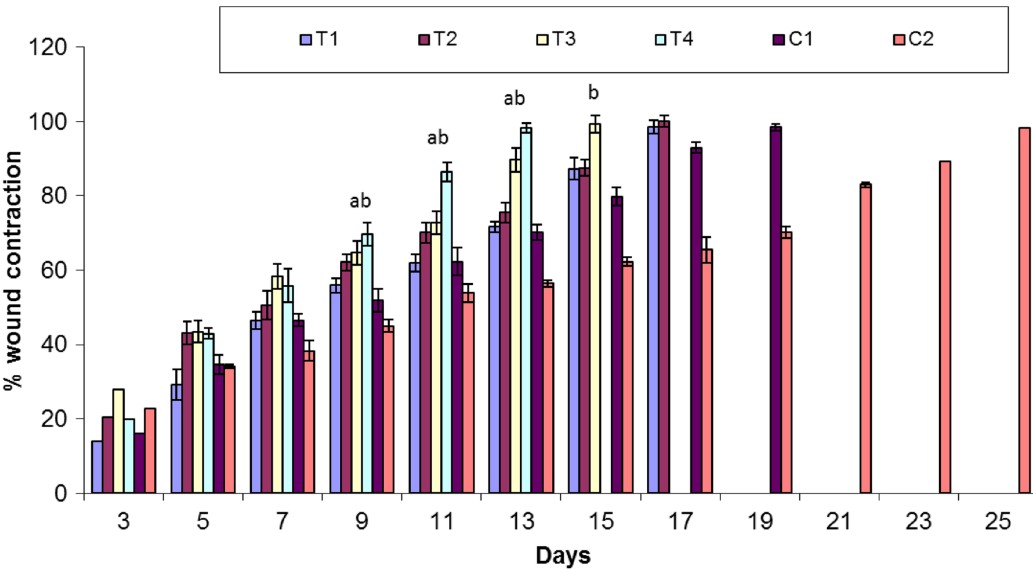

**Figure 3 Comparison of percent wound contraction between treatment and control groups.** Keys: C1 = Positive control (Polyfax); C2 = Negative control (Saline solution); T1 = 5% Sericin; T2 = 5% Sericin and 5% *Aloe vera* gel; T3 = 5% Fibroin; T4 = 5% Fibroin and 5% *Aloe vera* gel. 'a' indicates the significance difference between C2 and T3, 'b' indicates the significance difference between C2 and T4, Each bar represents the mean values and SEM of six replicates. Statistical icons: a, b = $p \le 0.05$.

group T2 (i.e., 5% sericin and 5% *Aloe vera* gel) showed adipose tissues and new epithelium and formation of new blood vessels and dermis at day 10. The histology of wound from group T3 (5% fibroin) showed an uneven epidermal surface. However, the epidermal surface became even and no ulceration was observed on day 10 (Fig. 4).

## DISCUSSION

In the present study, the potential of silk-based formulations to accelerate the wound healing process in diabetic mice was investigated. The results of this study indicated that silk sericin and fibroin when blended with *Aloe vera* gel quicken the healing process without causing any allergic reactions.

The wounds treated with 5% silk fibroin and 5% *Aloe vera* gel showed 85% healing in 11 days, however; wounds treated with 5% silk sericin and 5% *Aloe vera* gel showed 85% healing in 15 days. The results of wound healing treated with 5% silk sericin and 5% *Aloe vera* gel are comparable with findings of *Aramwit & Sangcakul (2007)* that 8% sericin cream significantly reduced wound healing time. Conversely, wounds treated with cream base healed in 15 days. Moreover, *Lamboni et al. (2015)* also reported that the incorporation of silk sericin in wound healing materials forms an exceptional biomaterial that stimulates re-epithelialization by improving the rate of migration, adhesion, growth of keratinocytes, fibroblasts and increased production of collagen at the wound site. In a clinical trial, (*Aramwit et al., 2013*) utilized 8% sericin combined with silver sulfadaizine cream (standard antibiotic cream) to treat open wounds caused by second-degree burns. Outcomes of the study showed that the average healing time of wounds was

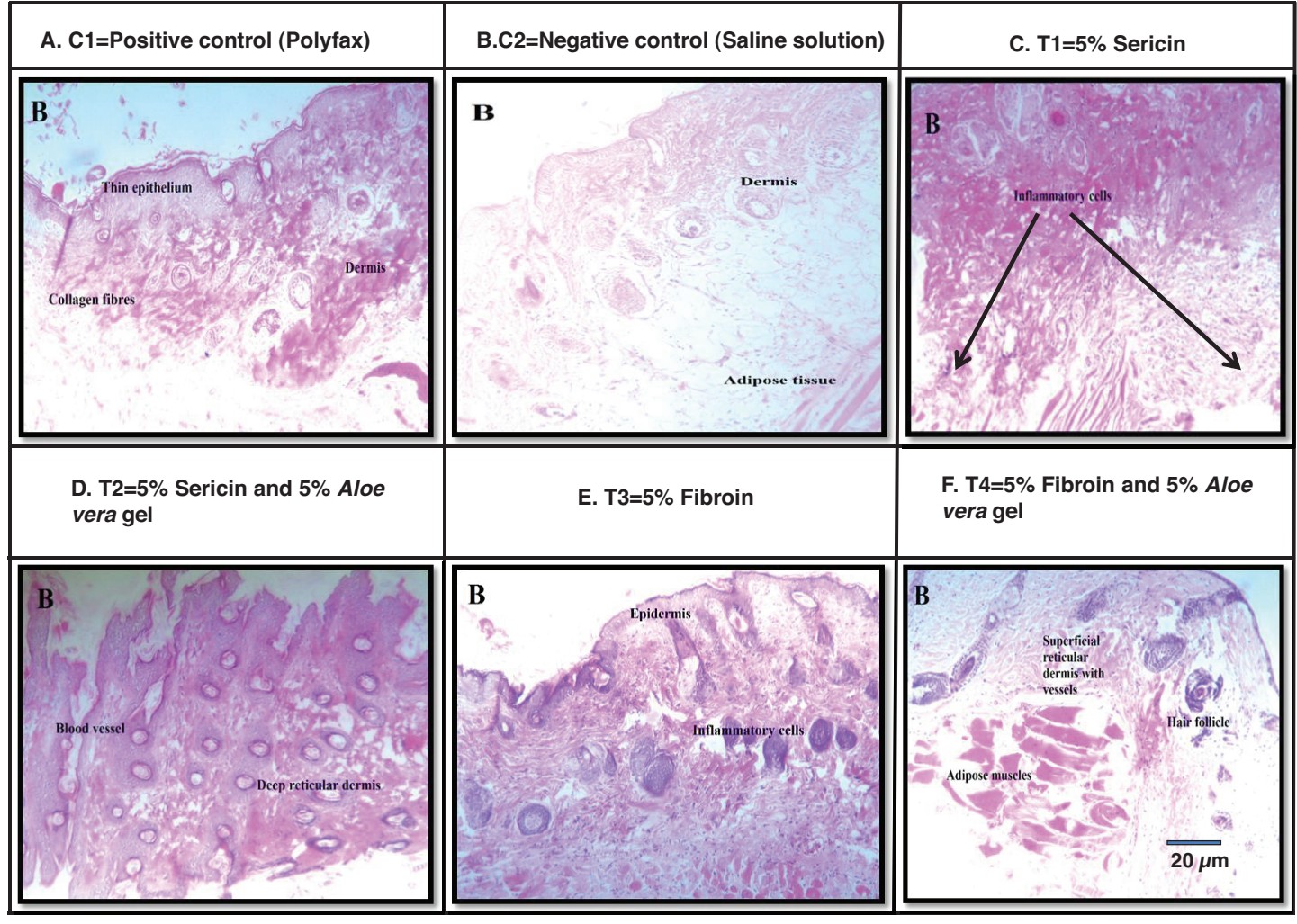

**Figure 4** H & E staining showing the histological changes in diabetic mice skin at post-wounding day 10 in different treatment groups. Magnifications of 10X. Scale bar = 100 μm. (A) C1 = Positive control (Polyfax); (B) C2 = Negative control (Saline solution); (C) T1 = 5% Sericin; (D) T2 = 5% Sericin and 5% *Aloe vera* gel; (E) T3 = 5% Fibroin; (F) T4 = 5% Fibroin and 5% *Aloe vera* gel.

significantly shorter in the treatment group compared to the control group (silver sulfadaizine without sericin).

Silk based films are considered safe and non-immunogenic biomaterial. The application of silk-based formulation on the skin does not affect serum profile since silk biofilm possess admirable biocompatibility with skin tissues. As it is infection-resistant in nature, it is regarded as an innovative wound coagulant biomaterial (*Padol et al., 2011*). The current study also indicated that silk proteins (sericin and fibroin) based formulations do not cause any skin irritation, infection, or allergy when applied topically on wounds of diabetic mice.

*Kanokpanont et al. (2013)* created a silk fibroin based bi-layered wound dressing. Silk fibroin woven fabric coated with wax was taken as a non-adhesive layer whereas the sponge composed of silk sericin and glutaraldehyde-cross linked silk fibroin/gelatin was

fabricated as a bioactive layer. Treatment of wounds with bi-layered wound dressings exhibited the greater potential of wound reduction, increased epithelialization, and collagen formation when compared with clinically available wound dressings. Hence this bi-layered wound dressing is considered as an excellent candidate for healing full-thickness skin wounds. Similarly, in another experiment *Baygar (2020)* investigated the synergistic effect of propolis and biogenic metallic nanoparticles in combination with silk sutures for biomedical use. It was observed that silk sutures coated with propolis and biogenic AgNPs showed potent antibacterial potential besides providing wound healing activity and biocompatibility. In the present study, sericin and fibroin were applied individually as well as in combination with *Aloe vera* gel on excision wounds in diabetic mice. The results indicated that 5% fibroin when mixed with 5% *Aloe vera* gel showed the best results among all treatment groups. Healing time till 85% wound contraction was reduced as compared to the control group (polyfax) 15–17 days. These findings suggest that silk can be amalgamated with other natural products like plant extracts to make it biogenic and to improve its medicinal properties.

*Aloe vera* is a medicinal plant that is widely being explored by scientists for its natural healing ability for skin and other delicate tissues (*Jadhav et al., 2020*). Earlier studies showed that one or more components of *Aloe vera* stimulate wound healing in different animal models (*Gallagher & Gray, 2003*). *Chithra, Sajithlal & Chandrakasan (1998)* analyzed the effects of *Aloe vera* gel on full thickness wounds in diabetic rats. Their results revealed that treatment with *Aloe vera* gel speeds up the wound healing process by increasing the rate of collagen synthesis, affecting fibroplasia and wound size reduction. In another study, *Maenthaisong et al. (2007)* evaluated the effectiveness of *Aloe vera* in burn wounds. *Aloe vera* was observed to increase the rate of re-epithelialization and reduce the wound healing period for burn wounds. The results of the current research have also showed that the treatment groups in which silk protein (fibroin) was combined with *Aloe vera* gel showed greater wound healing potential as compared to a positive control (polyfax). This combination of silk and plant extract was also observed to be most biocompatible as compared to other treatment groups because no inflammation or ulceration was observed on the skin of diabetic mice during the experiment.

## CONCLUSION

The results of this study suggests that silk based formulations can be utilized in wound healing materials because they are biocompatible, non-immunogenic and reduce wound healing time. This potential of silk-based formulations prepared in combination with *Aloe vera* gel has not previously been explored. Although the current research demonstrated the potential of silk derived formulations for wound healing in diabetic mice, the underlying molecular factors and events influencing wound healing are yet to be explored. Still, further studies need to be conducted to pinpoint how silk proteins influence the molecular events involved in the wound healing process. Improving wound healing treatments will improve the quality of life of diabetic patients suffering from chronic wounds along with a reduction in their health care costs.

### Funding

This work was supported by Higher Education Commission, Pakistan Technology Development Fund (No.TDF03-047). The funders had no role in study design, data collection and analysis, decision to publish, or preparation of the manuscript.

### Grant Disclosures

The following grant information was disclosed by the authors:
Higher Education Commission, Pakistan Technology Development Fund: TDF03-047.

### Competing Interests

The authors declare that they have no competing interests.

### Author Contributions

- Muniba Tariq conceived and designed the experiments, performed the experiments, analyzed the data, prepared figures and/or tables, authored or reviewed drafts of the paper, and approved the final draft.
- Hafiz Muhammad Tahir conceived and designed the experiments, performed the experiments, analyzed the data, prepared figures and/or tables, authored or reviewed drafts of the paper, and approved the final draft.
- Samima Asad Butt conceived and designed the experiments, performed the experiments, analyzed the data, prepared figures and/or tables, authored or reviewed drafts of the paper, and approved the final draft.
- Shaukat Ali conceived and designed the experiments, performed the experiments, analyzed the data, prepared figures and/or tables, authored or reviewed drafts of the paper, and approved the final draft.
- Asma Bashir Ahmad conceived and designed the experiments, performed the experiments, analyzed the data, prepared figures and/or tables, authored or reviewed drafts of the paper, and approved the final draft.
- Chand Raza conceived and designed the experiments, performed the experiments, analyzed the data, prepared figures and/or tables, authored or reviewed drafts of the paper, and approved the final draft.
- Muhammad Summer conceived and designed the experiments, performed the experiments, analyzed the data, prepared figures and/or tables, authored or reviewed drafts of the paper, and approved the final draft.
- Ali Hassan conceived and designed the experiments, performed the experiments, analyzed the data, prepared figures and/or tables, authored or reviewed drafts of the paper, and approved the final draft.
- Junaid Nadeem conceived and designed the experiments, performed the experiments, analyzed the data, prepared figures and/or tables, authored or reviewed drafts of the paper, and approved the final draft.

## Animal Ethics

The following information was supplied relating to ethical approvals (i.e., approving body and any reference numbers):

The Institutional Bioethics Committee at Government College University Lahore approved this research (No. GCU/IIB/21 dated: 08-01-2019).

## Data Availability

The raw calculations of percent wound contraction in all groups are available in a Supplemental File.

## Supplemental Information

Supplemental information for this article can be found online at http://dx.doi.org/10.7717/peerj.10232#supplemental-information.

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
