# Peer review of "Silk derived formulations for accelerated wound healing in diabetic mice"

_PeerJ, doi:10.7717/peerj.10232_

## Round 0.1 · original submission · Major Revisions

As you will see, there is a disparity between the two reviewers. While reviewer-2 raises some minor points, there are substantive concerns raised by reviewer-1, who did not recommend acceptance of the manuscript in this form. Therefore, I felt that you should be given an opportunity to address the issues raised by reviewer-1, in particular the comment regarding the model (point 1) is worrisome. You must address each of these issues systematically in any revision, which will also require to be re-reviewed.

The level of rigour must be increased throughout - note the comments regarding Figures 1, 2 and 3, and Table 3.

Please provide a detailed rebuttal to these comments if you decide to re-submit. However, you may feel upon reading these comments that another more specialised journal may be a better 'home' for this work.

Reviewer 1 ·

Basic reporting

The manuscript by Tariq et al entitled “Silk derived formulations for accelerated wound healing in diabetic mice” examines the potential of doping silk fibroin and sericin hydrogles with Aloe vera to generate a potential wound treatment product. The manuscript contains a range of data sets but fails to reach the minimum scientific rigour required by this journal. I have got the following specific comments that may help to further improve the manuscript:

Selection (many more problems):
(1) The in vivo study is flawed. Rodent wound healing is by contraction. In the absence of the splint model the data are misleading. Please see: Adv Wound Care (New Rochelle). 2013 May; 2(4): 142–148.
doi: 10.1089/wound.2012.0424
(2) Results: Figure 2. I am not convinced this is silk. This looks like salt crystals. Authors also mix up scales. SEM does not appear to be in the nanometer range as described in the body of the manuscript.
(3) Results: Figure 3. Duplication of images (left and right hand side). Top left figure panel: 102 nm is incorrect.
(4) Results: Table 3 needs to be converted into a bar chart.
(5) Discussion Line 256 following: Based on Table 3 not obvious how day 11 and 15 were selected and seems arbitrary.
(6) Mice: The wound area was not shaved properly. What was the sex of the animals?
(7) The vast majority of references are outdated. For example, the introduction cites the 2003 Altman review despite numerous timely and up-to-date reviews published over the past 5 years.
(8) Remove Figure 1 because it does not have any content.
(9) Figure scale bars are absent or too small.
(10) Figure 4. Overall appears sloppy: Too small, the annotations are not clear, scale bars are missing etc.
(11) Results: All figure legends need to be expanded to include more key information.
(12) The manuscript would benefit from English language editing (e.g. abstract “finest biocompatibility”; “wounds in control group (polyfax) were healed” etc.

Experimental design

Please see Basic Reporting

Validity of the findings

Please see Basic Reporting

Additional comments

Please see Basic Reporting

Reviewer 2 ·

Basic reporting

In general, language used is clear and professional English is used.
Sufficient background information is provided with proper citations except for some parts under methodology (shown in the manuscript as track changes).
Proper article structure is used with figures, tables and raw data.

Following comments are for further improvement
Keep a space between a numerical value and units
Additional reference is requested under methodology
Some figures need revisions (font size of labels, additional labels in Figure 4, comment on Figure 5 T4 day 11 panel)

Experimental design

Research question is well addressed
Ethical standards were maintained.
Experiments conducted have used accepted standards.

Following comments are for further improvement
Originality could be more emphasized- focusing on to wound healing in diabetes
One objective may be removed (shown in manuscript with track changes)
Comments-on extraction of sericin and measurement of wound contraction need to be addressed

Validity of the findings

In general results are supported with relevant figures and tables
Conclusions match with research question


Comment on figure 5 T4 panel need to be addressed (shown in manuscript with track changes)

Usefulness of tables 1 and 2 are not well documented

Additional comments

Comments are covered by 3 areas above

Annotated reviews are not available for download in order to protect the identity of reviewers who chose to remain anonymous.

---

## Round 0.2 · Minor Revisions

Thanks for your patience. One (re)-reviewer still felt strongly that we should reject. I felt this possibly harsh, so in the interest of fairness I ended up seeking further opinions on this article which has caused a little delay; I hope you will indulge this as the reviewers are helpful. They are positive and the suggestion of all three is that publication is warranted should you be able to make some minor amendments.

Please take careful note of the issues raised by reviewer-3 (SEM info, scale bars/clarity on the SEM figure and the labels on the H&E stains should be improved). Reviewer-5's comments need to be fully and carefully addressed, but these do not involve experimental work.

If you can clearly address these points and return the manuscript, I should be able to make a final decision without re-review fairly quickly.

Thanks for your understanding.

Reviewer 1 ·

Basic reporting

Addressing point 1 is beyond the scope of a revision. The model system is not robust and thus not of the required standard for Peer J.

Experimental design

See above.

Validity of the findings

n/a

Additional comments

Addressing point 1 is beyond the scope of a revision. The model system is not robust and thus not of the required standard for Peer J.

Reviewer 3 ·

Basic reporting

The article entitled "Silk derived formulations for accelerated wound healing in
diabetic mice” by Tariq et al is written in clear English. The language needs some improvement, but the formulated hypothesis and scientific method to address research question are clear and well structured.

Experimental design

In vivo study design is fairly solid with five treatment groups and a negative control. Moreover, given the ethical constrains, six animals per treatment group are convincing numbers "N" to conclude the rigor and strength of the findings based on statistical grounds. Method has sufficient detail and where aver possible well supported by relevant reference.

Validity of the findings

Although the silk proteins are known to induce beneficial wound healing effect however combination with medicinal plant extract in this current study has significantly improved wound healing in diabetic rodent model. Findings are strong and may eventually seek applications in pharma industry and medical sciences.

Additional comments

I reviewed the article entitled “Silk derived formulations for accelerated wound healing in
diabetic mice” by Tariq et al. In this manuscript, the authors have developed a new wound healing formulation by combining silk derived proteins, fibroin and sericin and aloe vera plant extract (gel). They then examined wound healing potential of this formulation in vivo on the chemically induced diabetic mouse model. The findings suggested that the combination of fibroin and Aloe vera gel provides beneficial wound healing effects as opposed to fibroin alone or a combination of sericin and aloe vera gel.

Although, as authors indicated, the wound healing effect conferred by the cocoon’s derived silk proteins is well known, however, the idea of combining silk protein with the extract of naturally existing medicinal plants and subsequently, testing the formulation on diabetic mouse model is very promising, which may eventually contribute to the fields of pharma industry and medical sciences.

As such the SEM images (fig 1 and fig 2) look suboptimal. I would suggest combining both figures as one with the same scale bar. If possible another confirmatory test could be done to show that the extracted proteins are indeed silk fibers. Since both proteins are available commercially, maybe a head to head western blot comparison or an in vitro cell based motility assay (doi: 10.1371/journal.pone.0042271) could be sufficient to show authenticity of the extracted proteins.

Authors should indicate statistical significance (p value) on the bars of figure 4 showing comparison of at least control versus silk protein and aloe vera formulations treated groups.

Labels within the H&E stained images are not readable, the authors should increase font size to make it visible.

Reviewer 4 ·

Basic reporting

The authors have successfully addressed all the queries raised by the other reviewers about the manuscript "Silk derived formulations for accelerated wound healing in diabetic mice". I have no further queries.

Experimental design

The experimental designs are looking good.

Validity of the findings

The conclusions are well stated and relevant to the current scenario of the field.

Additional comments

No further comments.

Reviewer 5 ·

Basic reporting

Interesting question, how different silk components act together with Aloe extracts in diabetic wound healing.
The manuscript is a revised version, which was clearly improved based on the reviewer’s comments.
Still the references are in inconsistent formats and partially incomplete eg.

Bouzghaya, S., Amri, M. and Homblé, F., 2020. Improvement of Diabetes Symptoms and Complications by an Aqueous Extract of Linum usitatissimum (L.) Seeds in Alloxan-Induced Diabetic Mice. Journal of Medicinal Food .
REV:Missing details
Cassinelli, C., Cascardo, G., Morra, M., Draghi, L., Motta, A. &Catapano, G., 2006. Physical-chemical and biological characterization of silk fibroin-coated porous membranes for medical applications.Int J Artif Organs., 29(9):881 .
REV:Only first page
Actabiomater ., 48:157-174.
REV: Unclear Journal name

Experimental design

Alloxan and streptozotocin are the most popular diabetogenic agents used for models, it is not mentioned why alloxan was used in this model and which type of diabetes does this represent.
A non-splinted excision model was used in mice. It should be mentioned what Polyfax is (antibiotic ointment)

Validity of the findings

I cannot find all the underlying data, only single examples of in vivo and histology figures are given.
Although the figures might have been improved, they are still of low quality. Fig 1 and 2 do not give a clear information, which comes also from the very limited figure legends. Quality of histology is limited.
It is unclear for which purpose the tables with MS results are shown, because there was no research question associated and also no discussion presented.

---

## Round 0.3 · accepted · Accept

Thank you for attending to these outstanding issues. I have now recommended acceptance.